# Robust Adversarial Reinforcement Learning in Stochastic Games via Sequence Modeling

**Xiaohang Tang**[*][†]
University College London

**Zhuowen Cheng**[*]
Independent Researcher

**Satyabrat Kumar**
University College London

## Abstract

The Transformer, a highly expressive architecture for sequence modeling, has recently been adapted to solve sequential decision-making, most notably through the Decision Transformer (DT), which learns policies by conditioning on desired returns. Yet, the adversarial robustness of reinforcement learning methods based on sequence modeling remains largely unexplored. Here we introduce the **Conservative Adversarially Robust Decision Transformer (CART)**, to our knowledge the first framework designed to enhance the robustness of DT in adversarial stochastic games. We formulate the interaction between the protagonist and the adversary at each stage as a stage game, where the payoff is defined as the expected maximum value over subsequent states, thereby explicitly incorporating stochastic state transitions. By conditioning Transformer policies on the NashQ value derived from these stage games, CART generates policy that are simultaneously less exploitable (adversarially robust) and conservative to transition uncertainty. Empirically, CART achieves more accurate minimax value estimation and consistently attains superior worst-case returns across a range of adversarial stochastic games.

## 1 Introduction

Transformer has established itself as a highly expressive architecture for sequence modeling, achieving state-of-the-art results in language modeling tasks (Vaswani et al., 2017). Building on this success, recent studies have recast reinforcement learning (RL) as a sequence modeling problem by representing states and actions as tokens, as exemplified by the Decision Transformer (DT) framework (Chen et al., 2021; Paster et al., 2022; Wu et al., 2024b; Tang et al., 2024a; Xu et al., 2023; Zheng et al., 2022). Yet, the capacity of sequence models to address robustness in sequential decision-making under adversarial perturbations remains largely underexplored. Prior efforts have predominantly targeted robustness to stochasticity in environments (Paster et al., 2022; Yang et al., 2022) or to corrupted data (Xu et al., 2024b; Takano et al., 2024), leaving the question of adversarial robustness in Decision Transformer unresolved.

Adversarially Robust Decision Transformer (ARDT) (Tang et al., 2024a) represents an early attempt to address adversarial robustness in offline RL via sequence modeling, where adversarial policy-distribution shifts arise. Tang et al. (2024a) show that the vanilla Decision Transformer (DT) is highly vulnerable under adversarial environments, owing to its formulation as goal-conditioned imitation learning—where achieving high returns during training may only attribute to the weakness of the behavioral adversarial policy rather than genuine robustness. ARDT mitigates this limitation by conditioning on minimax returns, thereby encouraging worst-case-aware policies. However, its applicability is restricted to adversarial RL with deterministic state transitions. In sequential stochastic games, where state transitions are inherently probabilistic, ARDT may exhibit over optimism, as

---

[*]Equal contribution.
[†]Corresponding author, `xiaohang.tang.20@ucl.ac.uk`.

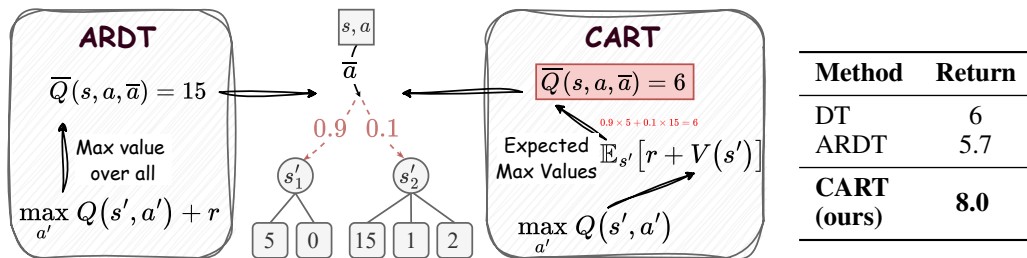

Figure 1: An illustrative example of stochastic game (detailed game setup in Figure 4). ARDT demonstrates overly optimism in estimating values of state and actions, regardless the *small probability to reach the desired state* $s_2'$. In contrast, our method CART addresses the stochasticity by assigning expected maximum values. Since the value estimation is more conservative and accurate, we call the proposed method Conservative ARDT (CART). We evaluate the returns against the optimal adversary across different methods in the right-hand-side table.

it conditions solely on minimax returns without accounting for transition probabilities of access to high-return subgames. The effect caused by this lack of conservatism has been shown in Figure 1.

We propose, to the best of our knowledge, the first method to enhance the adversarial robustness of the Decision Transformer in *stochastic* games, which we term the **Conservative Adversarially Robust Decision Transformer (CART)**. At each stage (i.e. time step $t$), interactions between the protagonist and adversary are formulated as a stage game, with the payoffs–the expected maximum value over subsequent states, thereby explicitly accounting for stochastic state transitions. To solve for the optimal minimax $Q$-values (NashQ) across all stages, we employ Expectile Regression combined with temporal-difference (TD) learning. In this way, by aligning minimax value estimation to the stochastic transitions, CART yields policies that are less exploitable and exhibit greater adversarial robustness when used for conditional sequence modeling.

## 2 Preliminary

In this section, we introduce our problem setup (sequential) stochastic game, and the base model Decision Transformer used to solve robust adversarial reinforcement learning *offline*.

### 2.1 Stochastic Game

Stochastic Game is defined by $(\mathcal{S}, \mathcal{A}, \bar{\mathcal{A}}, T, R)$, where $\mathcal{S}$ is the state space, $\mathcal{A}$ and $\bar{\mathcal{A}}$ are the protagonist and adversary action spaces, $R$ is the reward function, $T$ is a probability distribution representing the the transition kernel $s_{t+1} \sim T(\cdot|s_t, a_t, \bar{a}_t)$. At each step $t \leq H$, the players observe $s_t$ and choose actions $a_t, \bar{a}_t$, yielding reward $r_t = r(s_t, a_t, \bar{a}_t)$ for the protagonist and $-r_t$ for the adversary. A trajectory is $\tau = (s_t, a_t, \bar{a}_t, r_t)_{t=0}^H$, with return-to-go $\widehat{R}(\tau_{t:H}) = \sum_{t'=t}^H r_{t'}$. Policies $\pi$ and $\bar{\bar{\pi}}$ may depend on history. We restrict our setting in offline RL, where the agent cannot interact with the environment and instead learns from a dataset $\mathcal{D}$ of trajectories generated by behavioral policies $(\pi_{\mathcal{D}}, \bar{\bar{\pi}}_{\mathcal{D}})$. The objective is to learn a protagonist policy robust to an adaptive adversary who can observe the action of protagonist before making action: $(\pi^*, \bar{\pi}^*) = \max_\pi \min_{\bar{\pi}}, \mathbb{E}_{\tau \sim \rho^{\pi, \bar{\pi}}}[\sum_t r_t]$ with trajectory distribution $\rho^{\pi, \bar{\pi}}$.

### 2.2 Decision Transformer

Decision Transformer (DT) (Chen et al., 2021) can be formulated by reinforcement learning via supervised learning (RvS), aiming to learn a causal mapping from the state $s_t$ conditioned on the goal $z$ with a behavior-cloning loss. Denote return-to-go $\widehat{R}(\tau_{t:H}) = \sum_{t'=t}^H r_{t'}$, DT has loss function

$$\mathcal{L}_{\text{DT}}(\theta) = -\mathbb{E}_{\tau_{0:t-1}, s_t, a_t \sim \mathcal{D}, z=Q(s_t, a_t)}\Big[\log \pi_\theta(a_t \mid \tau_{0:t-1}, s_t, z)\Big], \tag{1}$$

where the condition $z$ is in training is set to $Q_{\text{DT}}(s_t, a_t) = \widehat{R}(\tau_{t:H})$, returns-to-go. In test-time when decoding the policy, the return can be maximized by setting a high target return $z$ such that the expected cumulative return can be maximized, i.e. $\forall t, \lim_{z \to \infty} \pi_\theta(a_t \mid \tau_{0:t-1}, s_t, z) \approx$

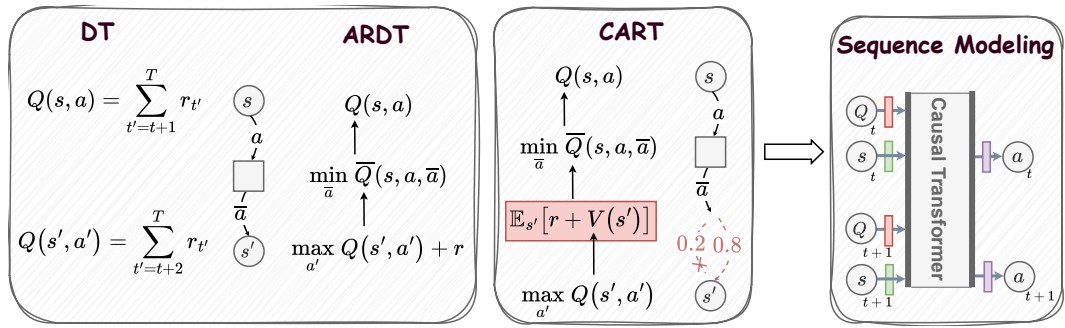

Figure 2: Return condition $z = Q(s, a)$ in training different DTs. In CART, additional function $V$ is added to address the stochasticity in state transition from $s$ to $s'$.

$\arg\max \mathbb{E}[\sum_t r_t]$. However, in adversarial reinforcement learning (RL), the high return might attribute to the weak behavioral policy of the adversary. Training by minimizing $\mathcal{L}_{\text{DT}}$ can lead to suboptimal policy which can only exploit weak adversary, lack of robustness to strong adversary. Adversarially Robust Decision Transformer (ARDT) (Tang et al., 2024a) set the goal $z$ to be the minimax returns rather than returns-to-go $\widehat{R}$ in DT to learn a worst-case-aware Transformer policy:

$$Q_{\text{ARDT}}(s_t, a_t) = \min_{\bar{a}_t} \max_{a_{t+1}} \min_{\bar{a}_{t+1}} \max_{a_{t+2}} \cdots \widehat{R}(\tau_{t:H}) \tag{2}$$

However, $Q_{\text{ARDT}}$ represents the accurate worst-case return only when the state transition function $T$ is deterministic. This assumption hinders ARDT to generate robust policy in stochastic games (Figure 1).

## 3 Method

In this section, we propose **Conservative Adversarially Robust decision Transformer (CART)**, a novel algorithm for addressing the adversarial robustness of Decision Transformer in stochastic games. Similar to the previous works, the critical part of CART is to conduct trajectory relabeling. We summarize different condition used for DT training across different methods in Figure 2.

To address the stochastic transition in CART, we leverage an additional state value function $V$. Following Nash $Q$-Learning (Hu and Wellman, 2003), we formulate a single stage of agents interaction and the state transition $(s, a, \bar{a}, s')$ as a stage game, where the protagonist makes action $a$ aiming to maximize the return, while the adversary sequentially takes action $\bar{a}$ after $a$ to minimize the return, and the state transition follows $s' \sim T(\cdot | s, a)$. The payoff function is defined by function $\bar{Q}(s, a, \bar{a})$ for all $a \in \mathcal{A}$ and $\bar{a} \in \bar{\mathcal{A}}$. Crucially, the payoff function has connection to the value functions in the next stage where

$$\bar{Q}(s, a, \bar{a}) = \mathbb{E}_{s' \sim T(\cdot | s, a)} [r + V(s')], \text{ and } V(s') = \max_{a'} Q(s', a'). \tag{3}$$

As defined in Equation 3, $V$ function represents the expected value by executing the optimal protagonist action in the next stage starting at $s'$. Accordingly, the stage-game payoff $\bar{Q}(s, a, \bar{a})$ should integrate over all possible subsequent states, evaluating the expected return under the corresponding state-transition probabilities. In this way, the stochasticity in state transition has been addressed.

As outlined in Section 2.1, the stage game proceeds sequentially, with the adversary observing the protagonist's action before responding. *The expected value of the stage-game solution (NashQ), which serves as the conditioning variable $z$ for DT training*, is defined as

$$Q_{\text{CART}}(s, a) = \min_{\bar{a}} Q(s, a, \bar{a}). \tag{4}$$

Importantly, if NashQ is computed at each stage and Equation 3 is satisfied, then every NashQ induces an adversarially robust policy that explicitly accounts for the stochasticity of state transitions.

We then introduce the practical algorithm to approximate the NashQ value. Notably, in reinforcement learning, training data are naturally structured as trajectories. This makes the $\min_{\bar{a}}$ or $\max_a$ inefficient as the operations are inaccurate until traversing all the data. We instead propose to employ Expectile Regression (Newey and Powell, 1987; Aigner et al., 1976) to approximate the operations, thereby

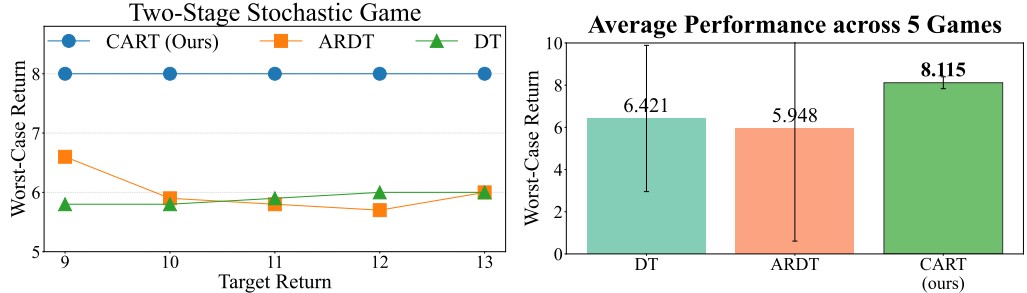

Figure 3: **LHS** demonstrates the worst-case return versus target return plot comparing the proposed CART algorithm against vanilla DT and ARDT, on our two-stage Stochastic Game. **RHS** represents the average performance Comparison among CART, ARDT, and DT across 5 synthetic adversarial stochastic games where we set high target return during decoding.

enabling $Q$-learning and the corresponding minimization (or maximization) to be performed jointly. In the beginning, we adopt MSE to learn the terminal state values. We then alternate the several loss optimization to solve the NashQ values in each stage games as follows.

We first learn the payoff of the stage-game by minimizing the TD error

$$\mathcal{L}(\bar{Q}) = \mathbb{E}_{(s,a,\bar{a},r,s')\sim\mathcal{D}}\big[\bar{Q}(s,a,\bar{a}) - V(s') - r\big] \tag{5}$$

to address the stochasticity in transition: $\arg\min_{\bar{Q}}\mathcal{L}(\bar{Q})(s,a) = \mathbb{E}_{s'\sim T(\cdot|s,a)}\big[r + V(s')\big]$). Subsequently, we estimate NashQ value $Q^*(s,a) = \min_{\bar{a}} Q(s,a,\bar{a})$ by minimizing the Expectile Regression (ER) objective:

$$\mathcal{L}(Q) = \mathbb{E}_{(s,a,\bar{a},r,s')\sim\mathcal{D}}\big[L_{\text{ER}}^{\alpha\to 0}(Q(s,a) - \bar{Q}(s,a,\bar{a}))\big], \tag{6}$$

where the ER objective is defined as $L_{\text{ER}}^{\alpha}(u) = \mathbb{E}\big[u|\alpha - \mathbf{1}(u > 0)| \cdot u^2\big]$. Finally, we approximate optimal state value function $V^*(s',a') = \max_{a'} Q(s',a')$ by minimizing ER

$$\mathcal{L}(V) = \mathbb{E}_{(s',a')\sim\mathcal{D}}\big[L_{\text{ER}}^{\alpha\to 1}(V(s') - Q(s',a'))\big]. \tag{7}$$

We alternatively optimizing the above three objectives and repeat for a large number of iterations. When the above algorithm converges, NashQ function $Q_{\text{CART}}$ converges to the value in Equation 4 is then used to train DT according to Equation 1.

## 4 Experiment

In this section, we assess the adversarial robustness of CART across a suite of synthetic stochastic games. Our evaluation is conducted in an offline setting, where training data are collected under a uniform behavioral policy. The central challenge lies in achieving robustness when learning from such unreliable training data, which constitutes the primary focus of our analysis.

**Experiment Setup.** We conduct experiments on synthetic Stochastic Game with stochastic transitions and adversarial actions, as described in Figure 4, 5 and 6. The Data is collected by employing uniformly random actions for both the protagonist and adversary, containing $10^5$ trajectories and encompassing all possible trajectories. In test time, we evaluate policy against adversary that is assumed to act optimally. So the metric used to indicate the adversarial robustness is worst-case return. We compare CART against ARDT and Decision Transformers. Following ARDT, our implementation of DT removes the condition on past adversarial tokens.

In Figure 3 on the LHS, we show the dynamics of varying target return in the illustrative two-stage stochastic game. CART achieves the highest worst-case return, conditioning on various large target return condition. While ARDT and DT under adversarial attack obtain lower return against the optimal adversary. In Figure 3 on the RHS, we summarize the average performance across 5 stochastic games, and CART obtains the highest worst-case return with the lowest variation.

ARDT can be misled by rare and high-payoff trajectories, overestimating the values of actions and neglecting true transition probabilities. This could lead to a lose of robustness at high target returns. Conversely, by jointly considering payoff and stochasticity in the state transition, CART focuses on feasible strategies that maximize worst-case expected return, yielding an adversarially robust policy.

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

## A  Experiment Setup

### A.1  Two-Stage Stochastic Game

In this second-stage Stochastic Game, when the protagonist selects action 0, the next state is stochastically determined based on the adversary's action. Suppose the adversary chooses $\bar{a}_0$, state 0 transitions to state 1 with 90% probability, where the protagonist can select among payoffs of 5, 5, and 0. Simultaneously, there is a 10% probability of transitioning to state 2, with payoffs of 15, 1, and 2. If the adversary chooses $\bar{a}_1$, state 0 will transition to state 2. For protagonist's actions 1 or 2, the payoff is 8 regardless of the adversary's choice. Figures below demonstrate 5 variants, including the original Stochastic Game, to test the robustness of CART.

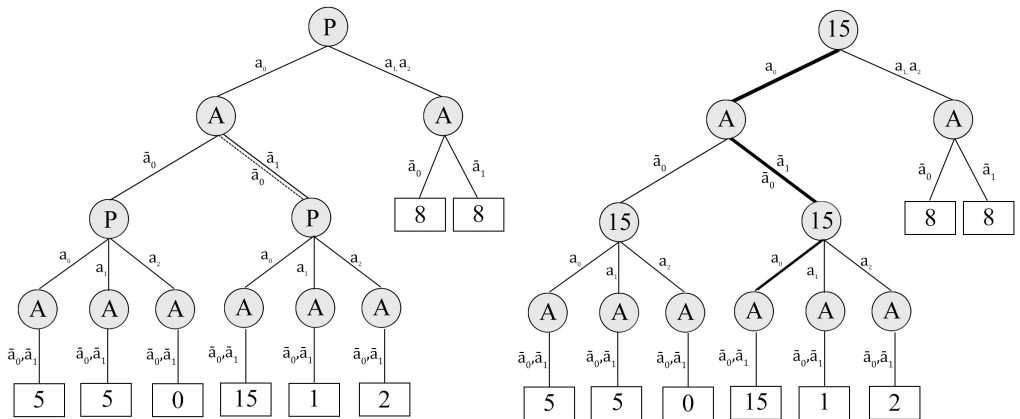

Figure 4: **LHS** presents the game with a target return of 8 where decision-maker P is confronted by Adversary A. In the worst-case scenario, if P chooses action $a_0$, A will respond with $\bar{a}_0$ to minimize P's expected payoff at the next state, and if P chooses $a_1$ or $a_2$, payoffs are independent of A's strategic behaviors. Consequently, the worst-case expected returns for actions $a_0$, $a_1$, and $a_2$ are 6, 8, and 8, respectively. Therefore, the robust choice of action for the decision-maker is $a_1$ or $a_2$. For training, the data are collected by running uniformly random behavior policy for long enough such that all the trajectories are covered. **RHS** represents the implementation of ARDT in this scenario. Due to the stochastic nature of the state transition, ARDT can be misled by rare, high-payoff events.

### A.2  Three-Stage Stochastic Game

This environment models a sequential interaction between two players across three stages. Each player has two possible actions at every stage. The game begins in an initial state. At each stage, the joint action determines a probabilistic transition to one of several possible next states. This process repeats for three stages in total. After the third stage, the system transitions into a terminal state where a fixed reward is given. This reward is observed only at the end of the episode. The randomness in transitions is demonstrated in Figure 8.

The experiment Results are demonstrated in Figure 7.

## B  Related Work

### B.1  Stochastic Game

Two-play game solving (i.e. approximating the Nash Equilibrium) has been well investigated in online learning (Zinkevich et al., 2007; Lanctot et al., 2009; Brown and Sandholm, 2019a,b; Erez et al., 2023; Tang et al., 2023, 2024b; Xu et al., 2024a), where online self-play is conducted to minimize the regret of actions, converging to the Nash Equilibrium with average policy. However,

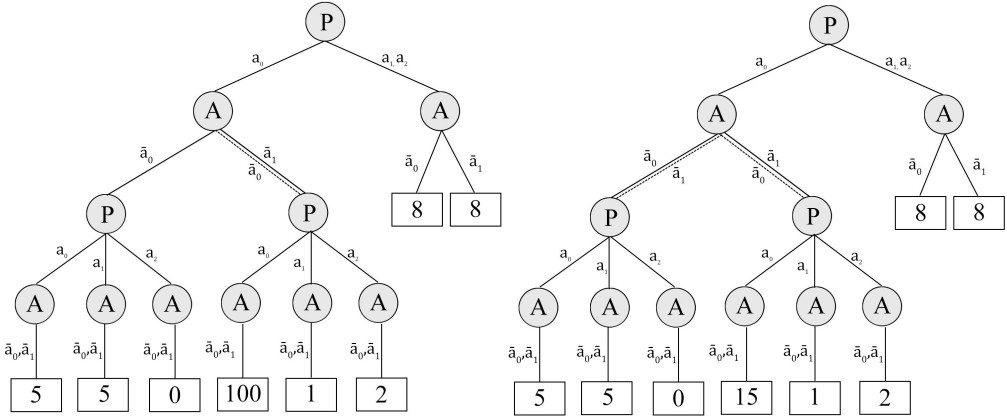

Figure 5: **LHS** demonstrates a variant of the two-stage Stochastic Game with a target return of 8 where the rare payoff is adjusted to be 100. **RHS** demonstrates a variant of the two-stage Stochastic Game with a target return of 8 when the Protagonist chooses action 0 at state 0 and the Adversary chooses action 1, the probability of transition to state 1 and 2 would be 20% and 80%, respectively.

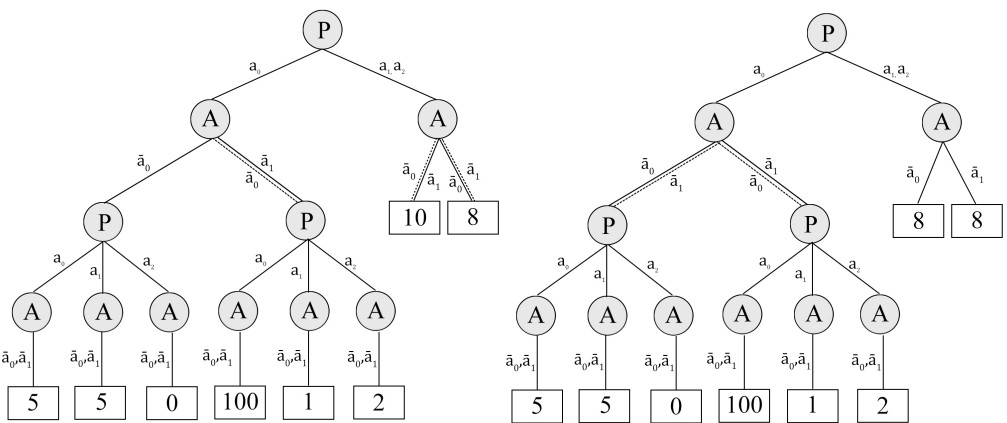

Figure 6: **LHS** demonstrates a variant of the two-stage Stochastic Game with a target return of 8.2 when the protagonist selects action 1 or action 2, the probability of receiving payoff 10 or 8 will vary based on the adversary's actions. **RHS** demonstrates a variant of the two-stage Stochastic Game with a target return of 8 by incorporating the properties of two variants in 4.

our work is in the setting of offline learning, where only a static dataset is provided and the online interactions are prohibited. We investigate how to learn non-exploitable policy from suboptimal dataset.

## B.2 Offline RL

Much of the prior work in offline RL has concentrated on stabilizing value estimation and learning robust policies when data is limited. Conservative Q-Learning (CQL) (Kumar et al., 2020) (Jiang et al., 2023) (Wu et al., 2024a) mitigates Q-value overestimation using a pessimistic objective. In contrast, Implicit Q-Learning (IQL) (Kostrikov et al., 2021) (Hansen-Estruch et al., 2023) achieves the value stabilization by learning an implicit value function via expectile regression. Following the inspiration of Implicit Q-Learning, (Tang et al., 2024a) demonstrates the adversarial robustness of Adversarially Robust Decision Transformer (ARDT) through minimax expectile regression in the static zero-sum games.

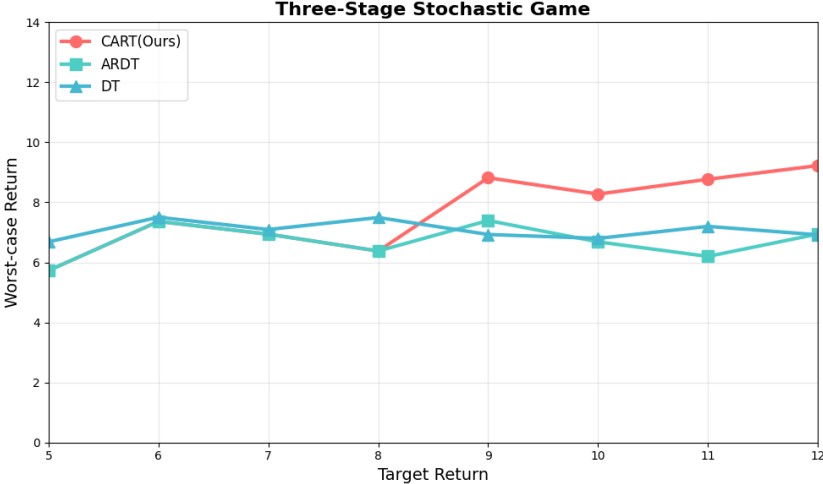

Figure 7: The figure demonstrates the performance comparison in the three-stage Stochastic Game transition setting among three algorithms.

However, the stochastic nature of games (i.e., randomness in state transitions) undermines the robustness of value function estimation. For instance, in ARDT, expectile regression inevitably reflects the stochasticity of environment dynamics and fails to perserve robustness in the stochastic games. Inspired by (Kostrikov et al., 2021; Tang et al., 2024a), we introduce an explicit value function and adapt ARDT to stabilize Q-function estimation in stochastic games, while preserving adversarial robustness in competitive environments.

### B.3 Decision Transformer

The Decision Transformer (Chen et al., 2021; Ma et al., 2023; Paster et al., 2022) reframes reinforcement learning as conditional sequence modeling, where a Transformer is trained on trajectories of return-to-go, states, and actions to autoregressively generate actions that achieve a desired return. Adversarially Robust Decision Transformer (ARDT) (Tang et al., 2024a) extends this framework by incorporating adversarial reasoning and max–min return objectives, thereby improving robustness in multi-agent and competitive settings. Other extensions of return-conditioned sequence models include the Trajectory Transformer (Janner et al., 2021), which captures trajectory stochasticity via latent variables; the Online Decision Transformer (Zheng et al., 2022), which integrates hybrid offline–online RL; the Skill Decision Transformer (Zhang et al., 2023), which leverages discrete skill representations to enhance cross-task generalization; and the Multi-Game Decision Transformer (Lee et al., 2022) and Prompt Decision Transformer (Xu et al., 2022; Yang and Xu, 2024), which support transfer learning and few-shot adaptation.

Building on this line of work, our Conservative Adversarially Robust Transformer (CART) leverages relabeled trajectories from offline datasets to train a decision transformer that remains effective and robust in stochastic games. By stabilizing value function estimation under transition randomness, CART strengthens the reliability of return-conditioned sequence modeling, ensuring more robust decision transformer training in competitive environments.

## C   Conclusion

In this paper, we introduce the Conservative Adversarially Robust Transformer (CART), a worst-case-aware offline RL algorithm that strengthens the adversarial robustness of the Decision Transformer in stochastic games. CART relabels trajectories using in-sample expected minimax returns-to-go, estimated via expectile regression and mean-squared error objectives. Empirical results on short-horizon stochastic games show that CART achieves improved robustness compared to both DT and ARDT. Future work can extend CART to more complex multi-agent and competitive environments, such as poker variants like Kuhn and Leduc Poker, where strategic reasoning under stochasticity

and adversarial interactions is crucial. This would test CART's ability to mitigate over-optimism, improve policy stability, and handle rare high-payoff events. Exploring larger-scale games and longer planning horizons could further evaluate its robustness and effectiveness.

**Stage 1 Transitions**

| Stage | P Action | A Action | Next Stage (probabilities) |
|-------|----------|----------|---------------------------|
| s0 | 0 | 0 | s1(0.7), s2(0.3) |
| s0 | 0 | 1 | s2(0.5), s3(0.5) |
| s0 | 1 | 0 | s1(0.6), s3(0.4) |
| s0 | 1 | 1 | s2(0.3), s4(0.7) |

**Stage 2 Transitions**

| Stage | Joint Action / Rule | Next Stage (probabilities) |
|-------|---------------------|---------------------------|
| s1 | P=0 | s5(0.6), s6(0.4) |
| s1 | P=1 | s5(0.4), s6(0.6) |
| s2 | A=0 | s5(0.7), s6(0.3) |
| s2 | A=1 | s5(0.3), s6(0.7) |
| s3 | (0,0) | s5(0.6), s6(0.4) |
| s3 | (0,1) | s5(0.4), s6(0.6) |
| s3 | (1,0) | s5(0.5), s6(0.5) |
| s3 | (1,1) | s5(0.3), s6(0.7) |
| s4 | any | s6 |

**Stage 3 Transitions (to Terminal)**

| Stage | P,A Action | Next Stage (probabilities) |
|-------|-----------|---------------------------|
| s5 | 0,0 | s7(0.6), s8(0.3), s9(0.1) |
| s5 | 0,1 | s7(0.5), s8(0.3), s9(0.2) |
| s5 | 1,0 | s7(0.3), s8(0.3), s9(0.4) |
| s5 | 1,1 | s7(0.2), s8(0.3), s9(0.5) |
| s6 | 0,0 | s7(0.5), s8(0.2), s9(0.3) |
| s6 | 0,1 | s7(0.4), s8(0.2), s9(0.4) |
| s6 | 1,0 | s7(0.25), s8(0.2), s9(0.55) |
| s6 | 1,1 | s7(0.2), s8(0.2), s9(0.6) |

**Terminal Rewards**

| Terminal Stage | Reward |
|----------------|--------|
| s7 | 25 |
| s8 | 10 |
| s9 | -15 |

Figure 8: The three-stage Stochastic Game setup.

