# OpenReview forum: "Robust Adversarial Reinforcement Learning in Stochastic Games via Sequence Modeling"
_NeurIPS.cc/2025/Workshop/Reliable_ML — NeurIPS 2025 - Reliable ML Workshop_

### Official Review · Reviewer_tPZN · 2025-09-13
**Robust Adversarial Reinforcement Learning in Stochastic Games via Sequence Modeling**

**Rating:** 8
**Confidence:** 4

**Review:**

Summary: The paper explores adversarially robust decision transformers, specifically by modifying the typical Q learning estimation with an interleaved adversary. The authors then demonstrate superior empirical performance in a sample task.

Strengths: The idea of modeling this as a two-stage game and fact that the computationally intractable "min max min ..." formulation of the objective could be reformulated as this three-stage iterative optimization procedure was very interesting. It took me a little bit to fully parse through the intuition behind the three-stage process, but the authors did a good job guiding the readers through the explanation. My understanding was that the estimation of $\overline{Q}$ and $V$ effectively follow the classical setup from Q learning, where we use optimistic value estimation, and that the value when playing against the adversary is then accounted for by the pessimism of $\alpha=0$ in the training of $Q$. The experiment was also a nice demonstration of this idea.

Weaknesses: Definitely not a deal-breaker by any means for this workshop, but as the authors extend their work, it would be interesting to consider larger experimental setups. Also, why do we have to do use the expectile loss for training $V$ instead of the hard argmax?

---

### Official Review · Reviewer_7v2j · 2025-09-18
**Conservative Adversarially Robust Decision Transformer: Interesting Idea, Yet Validation Remains Narrow**

**Rating:** 5
**Confidence:** 3

**Review:**

## Summary:
This paper studies the adversarial robustness of DT in stochastic games. While prior work (ARDT) incorporates minimax objectives, it assumes deterministic transitions and becomes overly optimistic when rare, high-reward events exist. The authors propose CART, which models each step as a stage game with payoffs defined as expected maximum values over possible next states. By estimating conservative NashQ values using expectile regression and conditioning DT on them, CART yields policies that are less exploitable. Experiments on synthetic stochastic games show CART achieves higher and more stable worst-case returns than DT and ARDT.

## Strengths. Novelty, rigor, empirical/theoretical quality, clarity, relevance to reliability with imperfect data.
1. By formulating protagonist–adversary interactions as stage games and introducing conservative NashQ conditioning, the approach effectively integrates robustness into sequence modeling.
2. The paper is well-structured with clear toy examples illustrating ARDT’s limitations, and experiments demonstrate that CART achieves higher and more stable worst-case returns than DT and ARDT, particularly under rare high-reward trajectories.

## Weaknesses / Limitations. Missing comparisons/ablations, unclear assumptions, proof gaps, failure modes, scope limits.
1. All experiments are restricted to small synthetic stochastic games (2–3 stage), without testing on more complex benchmarks, leaving practicality and scalability uncertain.
2. The method requires stage-wise NashQ estimation with repeated expectile regression, which may become prohibitively expensive in long-horizon or large-scale environments.
3. The paper does not analyze computational complexity or provide insights into the feasibility of applying CART to more realistic and high-dimensional settings.

## Ethics (if applicable). Note any concerns (about privacy, fairness, misuse, sensitive data use) and suggested mitigations.
N/A